



# Cold pools mediate mesoscale adjustments of trade-cumulus fields to changes in cloud-droplet number concentration

Pouriya Alinaghi[1], Fredrik Jansson[1], Daniel A. Blázquez[1], and Franziska Glassmeier[1]

[1]Department of Geoscience & Remote Sensing, Delft University of Technology, Delft, The Netherlands

**Correspondence:** Pouriya Alinaghi (p.alinaghi@tudelft.nl)

**Abstract.** The mesoscale self-organization of trade-cumulus cloud fields is a major cloud-climate uncertainty. Cold pools, i.e. pockets of cold, dense air resulting from rain evaporation, are a key mechanism in shaping these dynamics and are controlled by the large-scale forcing. We study the microphysical sensitivity of cloud-field self-organization through cold pools by varying cloud-droplet number concentration $N_c$ from 20 to 1000 /cm$^3$ in large-eddy simulations on large $154 \times 154$ km$^2$-domains.

We find that cold pools exhibit two distinct regimes of mesoscale self-organization. In very low $N_c$-conditions, cold pools transition from a stage where they are small and randomly distributed to forming large, long-lived structures that perpetuate due to the collisions of cold pools at their fronts. Under high-$N_c$ conditions, cold pools display strongly intermittent behaviour and interact with clouds through small, short-lived structures. While $N_c$ thus influences the number of cold pools and, in turn, mesoscale organization, cloud depth, and cloud albedo, we find its effect on cloud cover to be minimal. Comparing the

microphysical sensitivity of cold-pool-mediated mesoscale dynamics to the external, large-scale forcing shows that $N_c$ is as important as horizontal wind and large-scale subsidence for trade-cumulus albedo. Our results highlight that cold pools mediate adjustments of trade-cumulus cloud fields to changes in $N_c$. Such mesoscale adjustments need to be considered if we are to better constrain the effective aerosol forcing and cloud feedback in the trade-wind regime.

## 1   Introduction

Clouds play a crucial role in the climate system by modulating the Earth's energy budget through their interactions with radiation. Their net effect is to cool the planet by reflecting incoming solar radiation back into space (Stephens et al., 2012). Clouds are one of the most important sources of uncertainty in climate projections. Firstly, the cloud feedback is the most uncertain feedback to the anthropogenic forcing of the climate system, which is mainly due to the uncertain response of shallow clouds to climate change (Schneider et al., 2017; Nuijens and Siebesma, 2019; IPCC AR6, 2023). Secondly, the

complex interactions between clouds and aerosols lead to the process uncertainty that makes the effective radiative forcing due to aerosol-cloud interactions the most uncertain forcing in the climate system (IPCC AR6, 2023; Bellouin et al., 2020).

Aerosol perturbations change the concentration of cloud-condensation nuclei and in turn cloud-droplet number concentrations $N_c$. Assuming a fixed cloud-liquid-water path, increased $N_c$ results in a larger number of smaller cloud droplets, leading to a larger surface area to interact with radiation, and in turn increased cloud-optical depth and cloud albedo, known as *Twomey*

effect (Twomey, 1977). In addition to this quasi-immediate effect that can be considered to occur on the spatiotemporal scales



of individual cloud parcels, changes in $N_c$ can also propagate to larger scales. On the single-cloud scale, increased $N_c$ reduces the efficiency of collision-coalescence processes through smaller radii, decreasing the rain-formation efficiency, thereby delaying precipitation formation, known as *Albrecht* or *lifetime* effect (Albrecht, 1989). This delay in precipitation formation allows clouds to live longer and in turn get deeper, which in the end precipitate more intensely. Such effects can lead to an internal re-organization on the cloud field or mesoscale. They can be considered a form of self-organization because they are not prescribed by a large-scale forcing. In addition to delayed precipitation formation, increased $N_c$ has also been described to affect entrainment rates with effects on meso-timescales (Glassmeier et al., 2021).

For trade-cumulus cloud fields, large-eddy simulations (LESs) were first employed on small domains ($6.4{\times}6.4\text{-}12.8{\times}12.8$ km$^2$) to investigate the response of shallow cumuli to aerosol perturbations (Xue et al., 2008; Zuidema et al., 2008). A decade later, LESs on larger domains (Seifert et al., 2015; Yamaguchi et al., 2019) showed that the compensating internal adjusting processes as proposed by Stevens and Feingold (2009) occur on the scales of $50 \times 50$ km$^2$ cumulus cloud fields, which is far beyond the scale of an individual cloud. While not their focus, these studies give a clear indication of cold-pool activity. Cold pools are pockets of cold, dense air resulting from downdrafts associated with rain evaporation. When these downdrafts reach the surface, cold pools spread outward in a circular pattern. Studies across various regimes—stratocumulus, shallow, and deep convection—indicate that cold-pool boundaries feature strong moist updrafts, which trigger cloud formation along their edges, forming cloud rings often visible in satellite imagery (Xue et al., 2008; Savic-Jovcic and Stevens, 2008; Zuidema et al., 2012; Böing et al., 2012; Jeevanjee and Romps, 2013; Schlemmer and Hohenegger, 2014; Langhans and Romps, 2015; Torri et al., 2015; Drager and van den Heever, 2017; Zuidema et al., 2017; Haerter and Schlemmer, 2018; Helfer and Nuijens, 2021; Lochbihler et al., 2021; Vogel et al., 2021; Touzé-Peiffer et al., 2022). When cold pools form in close proximity, their boundaries can collide (Torri and Kuang, 2019), which intensifies the next convective event. Such cold-pool interactions through collisions implement self-organization, as has been conceptually modeled for both open-cell stratocumulus (Glassmeier and Feingold, 2017) and deep convective regimes (Haerter et al., 2019; Nissen and Haerter, 2021).

Shallow cumuli in the trades are frequently precipitating (Nuijens et al., 2009; Snodgrass et al., 2009; Radtke et al., 2022), leading to the frequent presence of cold pools in the trade-wind regime (Zuidema et al., 2012; Vogel et al., 2021; Touzé-Peiffer et al., 2022). The size and frequency of occurrence of cold pools covary with the mesoscale organization of trade-cumulus clouds (Vogel et al., 2021). Even under large-scale conditions that are invariant in time and space, LES studies show that trade-cumulus cold pools self-organize and in turn pattern trade-cumulus fields into arc-shape structures (Seifert and Heus, 2013; Vogel et al., 2016). By generating strong moist updrafts at their fronts, cold pools affect and interact with clouds (Zuidema et al., 2012; Li et al., 2014; Vogel et al., 2021; Alinaghi et al., 2024c). On large-domain LESs of the trade-wind regime, Alinaghi et al. (2024c) recently showed that the cold-pool-cloud interaction expresses itself in the form of structures resembling shallow squall lines. Thus, cold pools are coupled to clouds through shallow circulations at the mesoscales which were frequently observed in the trades (George et al., 2023) and affect cloudiness (Vogel et al., 2022; Janssens et al., 2023; Alinaghi et al., 2024c).

The mesoscale dynamics of trade cumulus are typically discussed in the context of cloud feedback. The trade-cumulus feedback has long been a large source of uncertainty in climate projections (Cesana and Del Genio, 2021; Myers et al., 2021).





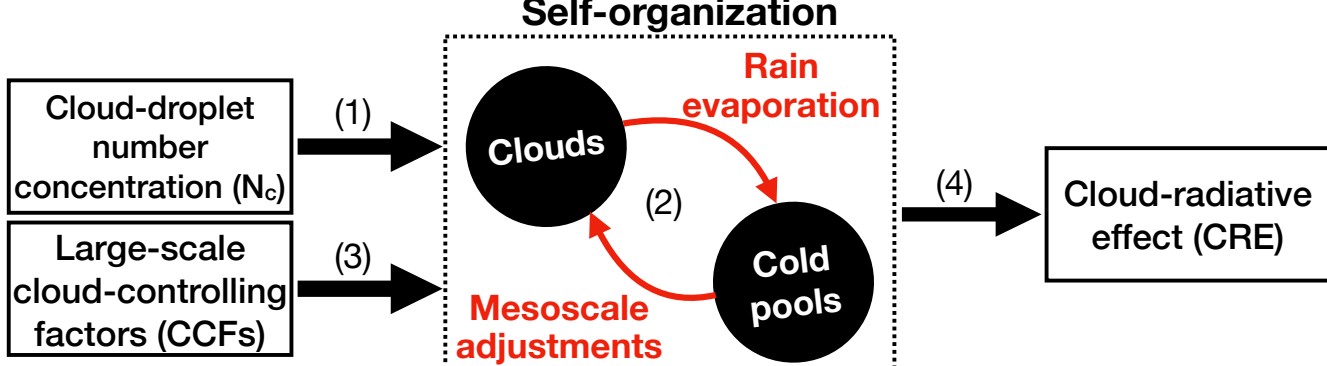

**Figure 1. Conceptual picture of the study.** The diagram summarizes how (link 1) cloud fields respond to changes in $N_c$, and how (link 2) cold pools feed back to clouds (self-organization). The diagram further shows that the trade-cumulus system is forced by (link 3) the large-scale cloud-controlling factors (CCFs) whose relative importance for (link 4) the cloud-radiative effect will be quantified compared to $N_c$.

Trade-cumulus fields pattern into structures at the mesoscales that influence cloud-radiative effect (Bony et al., 2020; Alinaghi et al., 2024a; Denby, 2023). Therefore, it is crucial to explore processes through which these clouds organize and how these processes respond to the variations in large-scale cloud-controlling factors (CCFs). By employing a large ensemble of LESs, the *Cloud Botany* ensemble (Jansson et al., 2023), Alinaghi et al. (2024c) illustrated that cold pools in the trades are strongly
controlled by the variations in the large-scale external CCFs. They particularly quantified the relative importance of CCFs with respect to each other. Additionally, diurnality in insolation, which acts as a time-varying CCF, was shown to strongly control the temporal evolution of cold pools throughout the entire *Cloud Botany* ensemble (Alinaghi et al., 2024c).

     Given the direct impact of $N_c$ on precipitation formation, it is expected that trade-cumulus cold pools respond to variations in $N_c$ (Fig. 1, link 1), and thereby to feed back to cloud fields at the mesoscales (Fig. 1, link 2). Despite the previously investigated
sensitivity of shallow cold pools to microphysics schemes (Li et al., 2015), the response of shallow cold pools to $N_c$ has not been directly explored and quantified. Furthermore, it is unknown how such $N_c$ variations change the interplay between cold pools and clouds in the trade-wind regime. This study explores this response by performing large-domain, large-eddy simulations in which we only vary cloud-droplet number concentrations $N_c$ from 20 to 1000 /cm$^3$. The newly added dimension of variability in $N_c$ here also enables us to systematically investigate the relative importance of $N_c$ compared to the other CCFs (Fig. 1, link
3) for cold pools and the radiative effect of clouds in the trade-wind regime (Fig. 1, link 4). Hence, our work serves as a step towards understanding the significance of the aerosol forcing in comparison to the trade cumulus feedback and exploring the corresponding role of mesoscale dynamics.

     This paper is structured as follows. Based on simulations discussed in section 2, we first investigate how trade-cumulus cold pools respond to $N_c$ variability (Fig. 1, link 1; sections 3.1, 3.2) and how this response shapes the mesoscale organization of
clouds (Fig. 1, link 2; section 3.2). Second, we explore how the diurnal cycle in insolation, as a time-varying CCF (Fig. 1, link



3), controls the evolution and response of cold pools to $N_c$ (section 3.3). Next, we investigate the implications of our results for the cloud-field adjustments to $N_c$ (Fig. 1, links 1, 2; section 3.4). Finally, we compare the effect $N_c$ on cloud-field properties and radiative effects to that of large-scale external CCFs (Fig. 1, links 3, 4; section 3.5). Conclusions are presented in section 4.

## 2   Data and Methods

We perform large-eddy simulations (LESs) with the Dutch Atmospheric LES (DALES) model over domains of 153.6 × 153.6 km$^2$, featuring a horizontal resolution of 100 m and a vertical resolution of about 20 m. All simulations are forced by the same large-scale CCFs. These follow the central reference simulation of the *Cloud Botany* ensemble (Jansson et al., 2023), which corresponds to the mean large-scale conditions of the winter trades as derived from the ERA5 reanalysis data (Hersbach et al., 2020). The corresponding profiles, which are also used for initialization are shown in Fig. 2. Moreover, all simulations feature the same horizontal tendencies of cooling and drying through advection as shown in the *Cloud Botany* paper (Jansson et al., 2023, their Fig. 3). Most of the simulations feature diurnality in the solar incoming radiation, while all other CCFs are fixed in time. Thus, the variability and evolution in the simulations are driven by the interaction between the components of the system, allowing the study of processes via which the system self-organizes. For more details on the design of the *Cloud Botany* simulations, including the selection of parameters for the large-scale forcing, refer to Jansson et al. (2023).

Simulations utilize the two-moment cloud-microphysics scheme of Seifert and Beheng (2001) with a constant cloud-droplet number concentration $N_c$. We conduct six, 72-hour simulations with varying $N_c$ values in the set $\{20, 50, 70, 100, 200, 1000\}$ /cm$^3$. The selected range of $N_c$ variability, from 20 to 100, is similar to that used by Seifert et al. (2015), which is based on observations of the trades (Colón-Robles et al., 2006; Gerber et al., 2008; Hudson and Noble, 2014). We also included $N_c$ values of 200 and 1000, as recent observations from the EUREC[4]A field campaign (Bony et al., 2017; Stevens et al., 2021) report $N_c$ values as high as 1000/cm$^3$, primarily due to the presence of dust (see Fig. 9 in Quinn et al., 2021, and Figs. 9 and 10 in Bony et al., 2022).

Figure 3 visualizes that all these simulations start from a homogeneous non-cloudy state and develop into randomly distributed cumulus clouds. Afterward, clouds self-aggregate due to the presence of self-reinforcing shallow mesoscale overturning circulations (Bretherton and Blossey, 2017; Narenpitak et al., 2021; Janssens et al., 2023). As clouds aggregate, they deepen and eventually start to precipitate, leading to the presence of mesoscale arc-like structures, indicating the presence of cold pools in the field.

Alinaghi et al. (2024c) showed that the evolution of cold pools across the entire *Botany* ensemble is strongly controlled by the diurnality of insolation, mirroring observations of the trades (Vial et al., 2021; Vogel et al., 2021). To investigate how the evolution of cold pools is affected by variations in $N_c$ independently of the diurnal cycle, we switch off the diurnality in insolation. To this end, we re-run simulations with $N_c$ values of 20, 70, and 1000 /cm$^3$, while keeping the solar zenith angle time-invariant, ensuring that the total incoming solar radiation over the entire 24-hour period is equal to that of the simulations with the diurnal cycle (Alinaghi et al., 2024c, their section 3.2).



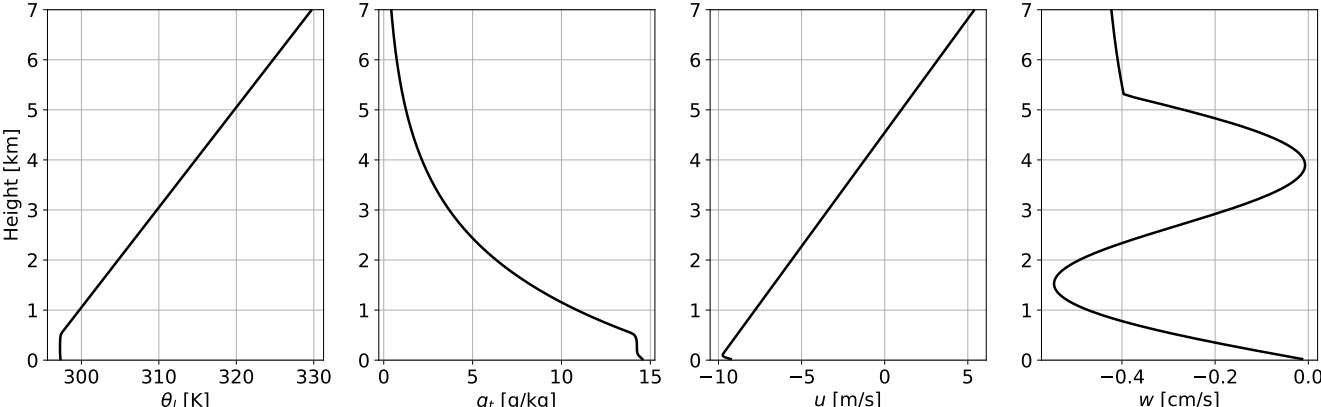

**Figure 2. Large-scale and initial conditions of the simulation** for potential temperature $\theta_l$, total moisture $q_t$, horizontal wind $u$, and updraft $w$ following the assumptions of the *Cloud Botany* ensemble (Jansson et al., 2023) with parameter values of sea-surface (potential) temperature $\theta_{l0} = 299$ K, near-surface wind speed $u_0 = -10.6$ m/s, moisture scale-height $h_{qt} = 1810$ m, temperature lapse-rate $\Gamma = 5$ K/km, large-scale vertical velocity variability $w_1 = 0.0393$ cm/s, and the horizontal wind shear $u_z = 0.0022$ (m/s)/m.

To diagnose cold pools, we use the 2D outputs of the mixed-layer height $h_{mix}$, as $h_{mix}$ has been shown to be a reliable
indicator for trade-cumulus cold pools in both models (Rochetin et al., 2021) and observations (Touzé-Peiffer et al., 2022). We identify cold pools following Alinaghi et al. (2024c): for each cloud field, we find the mode and the upper boundary (99th percentile) of an assumed symmetric probability density function (PDF) of $h_{mix}$ that would have been observed in the absence of cold pools. The lower boundary of $h_{mix}$ is determined by subtracting the difference between the upper boundary and the mode from the mode. Cold pools are identified where $h_{mix}$ is smaller than the lower boundary of $h_{mix}$ (see Fig. 3 in Alinaghi
et al. (2024c)). In essence, this method identifies cold pools where $h_{mix}$ is relatively shallower compared to other parts of the field. Alinaghi et al. (2024c) showed that the response of cold pools to CCFs is not sensitive to the details of the cold-pool diagnosis performed with this method.

## 3 Results and Discussion

### 3.1 Cloud-droplet number concentration affects the spatial and temporal properties of trade-cumulus cold pools

In this section, we investigate how the spatial and temporal properties of cold pools are influenced by $N_c$ (Fig. 1, link 1, 2). As cold pools result from rain-evaporation (Fig. 1, link 2), we first examine how clouds and rain respond to $N_c$ in simulations without diurnal cycle, which feature $N_c$ values of 20, 70, and 1000 /cm$^3$. According to theory (Albrecht, 1989) and previous LES studies (Seifert et al., 2015; Yamaguchi et al., 2019), increased $N_c$ reduces the efficiency of auto-conversion, delaying rain formation and allowing clouds to deepen and persist longer. Consistent with this, increased $N_c$ leads to the accumulation of





**Figure 3. Cloud-field albedo examples.** The figures show how cloud fields develop in our LES simulations featuring the diurnal cycle of insolation with $N_c$ of 20 (1st row), 50 (2nd row), 70 (3rd row), 100 (4th row), 200 (5th row), and 1000 /cm$^3$ (6th row), at hours 15 (1st column), 30 (2nd column), 40 (3rd column), and 50 (4th column) after the start of simulations.





**Figure 4. Time series of clouds, rain, and spatial properties of cold pools in simulations without the diurnal cycle.** Panels (a-d) show time series of $\mathcal{L}$, $\mathcal{R}$, $n_{cp}$, and $s_{cp}$ for simulations without the diurnal cycle of solar incoming radiation. Panels (e-h) show the response of $\mathcal{L}$, $\mathcal{R}$, $n_{cp}$, and $s_{cp}$ to $N_c$. The results from Yamaguchi et al. (2019) averaged over the last 20 hours of their simulations are shown in orange. The results from Seifert et al. (2015) for their near-equilibrium state are shown in blue. Dashed lines are added as a visual guide where a metric is sensitive to $N_c$. Note that the reason for the cold pools in simulation $N_c = 70$ /cm$^3$ forming earlier than $N_c = 20$ /cm$^3$ is that cold pools featuring a diameter of <5 km is not considered to be part of the cold-pool mask. Time series of cold-pool number and size are slightly smoothed using Gaussian filtering.

liquid water $\mathcal{L}$ (Fig. 4a), eventually resulting in the production of more intense rain $\mathcal{R}$ (Fig. 4b). Furthermore, the amplitude of fluctuations of $\mathcal{L}$ and $\mathcal{R}$ increases with increasing $N_c$, while their frequency decreases (Figs. 4a,b).

We quantitatively compare our results to those of Yamaguchi et al. (2019) and Seifert et al. (2015) (Figs. 4e,f). Results of Yamaguchi et al. (2019) were obtained on ten times smaller domains but using a microphysics scheme with prescribed aerosol and prognostic cloud-droplet number concentrations. The latter converges to a certain value after approximately 20

hours (Yamaguchi et al., 2019, their Fig. 5e). We therefore compare our fixed-$N_c$ results to averages of the last 20 hours of their





simulations. The systematic difference between the values of cloud-field properties is expected due to differences in the large-scale cloud-controlling factors; notably, their geostrophic wind speed is 60% smaller than that of our simulations (Yamaguchi et al., 2019, their Table 1). Similarly, we present the results from Seifert et al. (2015), which feature a domain size similar to that of Yamaguchi et al. (2019) but fixed $N_c$ as in our simulations. We selected their simulations with interactive radiation and

prescribed large-scale advective cooling to match our setup as closely as possible. It is worth noting that the cloud fields in Seifert et al. (2015) have larger cloud- and rain-water paths (Figs. 4e,f), which we mainly attribute to their larger meridional geostrophic wind.

Consistent with Yamaguchi et al. (2019) and Seifert et al. (2015), increased $N_c$ leads to an increased domain-mean liquid-water path, $\mathcal{L}$ (Fig. 4e). As shown by the error bars, the temporal variance in $\mathcal{L}$ increases in response to increased $N_c$ in

our simulations. In addition, Fig. 4f illustrates that the domain-mean rain-water path, $\mathcal{R}$, decreases with increasing $N_c$ in all studies. Similar to $\mathcal{L}$, the temporal variations in $\mathcal{R}$ increase with increasing $N_c$ in our simulations. This is in contrast to the small-domain LESs of Yamaguchi et al. (2019, $50 \times 50$ km$^2$) and Dagan et al. (2018, $12 \times 12 - 50 \times 50$ km$^2$), where increased $N_c$ was found to reduce the amplitude of fluctuations in the time series of $\mathcal{R}$. The temporal variance of liquid- and rain-water path in simulations of Seifert et al. (2015) do not show a systematic response to $N_c$. The response of the temporal variations of

cloud-field properties to $N_c$ thus appears markedly different in our large domains.

The temporal variations of cold-pool characteristics follow those of the rain-water path, $\mathcal{R}$ (Figs. 4b-d). Notably, in the simulation with $N_c = 20$/cm$^3$, once cold pools form around hour 30, they grow until around hour 36 and persist until the end of the simulation. In contrast, cold pools in the simulation with $N_c = 1000$/cm$^3$ form, develop, reach a maximum, decay, and completely vanish. Therefore, increased $N_c$ enhances the intermittency in the evolution of cold pools. Averaged over the last

day, increased $N_c$ leads to a smaller number of cold pools, $n_{cp}$ (Fig. 4g), while the domain-mean size of cold pools, $s_{cp}$, shows a muted response to $N_c$ variations (Fig. 4h).

## 3.2 Cold-pool evolution shows two distinct mesoscale behaviours for low and high cloud-droplet number concentrations

To understand the difference in cold-pool dynamics for different cloud-droplet number concentrations, Fig. 5 contrasts the

extreme $N_c$ cases. Consistent with observations (Zuidema et al., 2012; Vogel et al., 2021), both have in common that cold pools alter the spatial pattern of moisture in the sub-cloud layer. Cold pools are characterized by relatively dry air inside but relatively moist air at their fronts. Cold pools further modify the spatial pattern of horizontal velocity in the sub-cloud layer, thereby altering the spatial pattern of convergence and, in turn, vertical velocity. Cold pools feature large values of downward vertical velocity inside but strong updrafts at their fronts. Therefore, cold-pool fronts actively contribute to moist convergence

and cloud formation. In contrast, the downward motions inside cold pools can potentially suppress convection and lead to cloud-free regions.

Despite the similar evolution of individual cold pools independent of $N_c$, there are notable differences in the collective, i.e. self-organization, behavior. In the low-$N_c$ simulation (i.e., $N_c = 20$/cm$^3$), cold pools tend to form in close proximity. As their fronts converge, they trigger the formation of new cold pools at their collision points. This triggering mechanism occurs



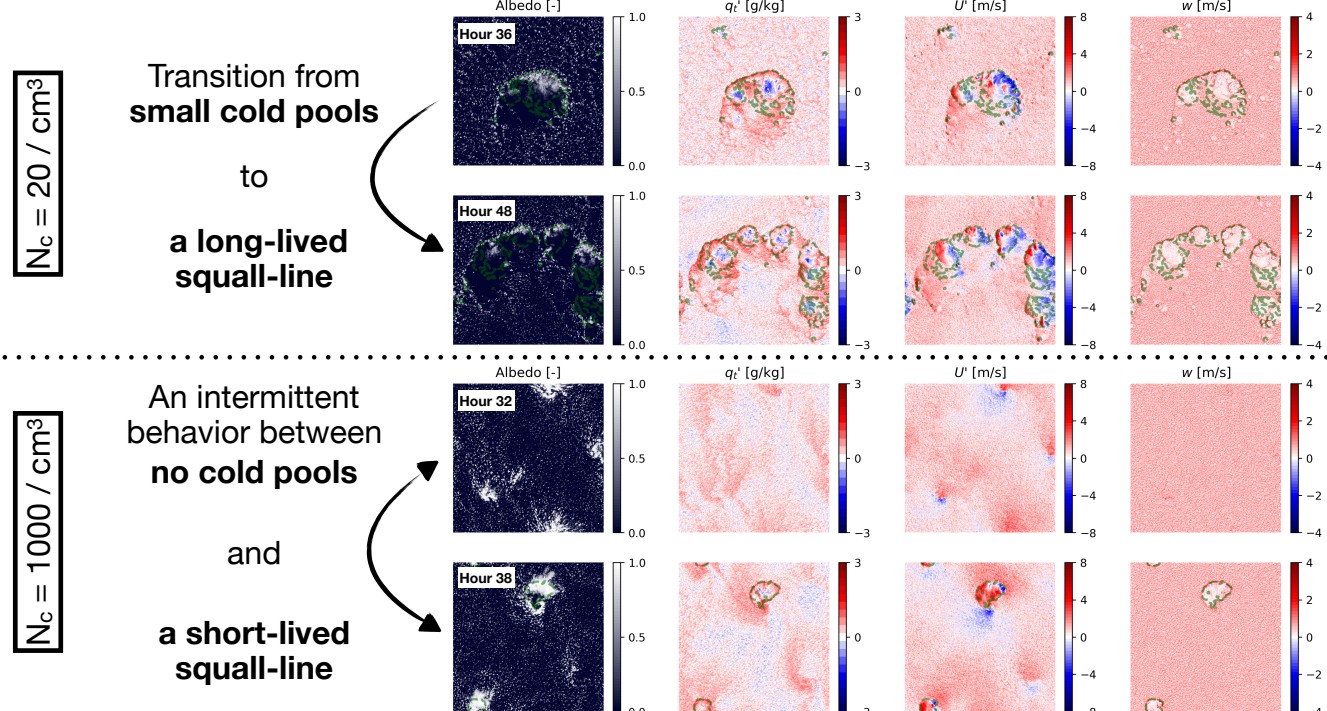

**Figure 5. Effects of cold pools on the organization of the (sub-)cloud layer properties for the simulations without diurnal cycle and with $N_c = 20$/cm$^3$ and $N_c = 1000$/cm$^3$.** For each condition, columns 1-4 indicate the 2D top-views of cloud albedo, total moisture anomalies ($q'_t$) at the 200-m level, the horizontal wind speed anomalies ($U'$) at the first level of the model, and the vertical velocity ($w$) at the 200-m height. The green dashed contour line marks the cold-pool boundaries quantified from the mixed-layer height fields. In the $U'$ fields, red indicates that cold pools accelerate the wind, while blue indicates they decelerate it.

due to (i) the collision of anomalously moist cold-pool fronts, which mechanically forces the moist air upward due to mass conservation, and (ii) the high efficiency of rain formation, where a small amount of cloud water quickly turns into rainwater, leading to the formation of a new cold pool at the collision point. Consequently, the small, space-filling cold pools at hours 30-36 collide and transition into a stage where they become organized into a large front, sustained by the interaction (collision) of its cold pools at its fronts (see Fig. 5, $N_c = 20$/cm$^3$, hour 48). This is consistent with the time series of cold-pool number $n_{cp}$ and size $s_{cp}$ shown in Figs. 4c,d, which show that after around hour 38, the metrics of cold pools stabilize: $s_{cp}$ remains around 15±5 km, and $n_{cp}$ around 6±4. This behaviour of cold pools resembles the mathematical toy model of colliding circle cold pools presented by Nissen and Haerter (2021, their Fig. 5), which demonstrates (i) similar transitions from randomly distributed cold pools to a band-like structure, and (ii) that cold-pool collision is the key mechanism for the self-organization of the system. Notably, their model was motivated by cold pools in the regime of deep convection.

In contrast, cold pools do not interact as readily in the high-$N_c$ simulation (i.e., $N_c = 1000$/cm$^3$). This is attributable to two factors. First, the efficiency of rain formation and, consequently, cold-pool formation, is significantly lower in high-$N_c$ than in





low-$N_c$ simulations. Second, cold pools form at greater distances from each other. We speculate that this distance is determined by the horizontal length scale of the self-reinforcing shallow circulations that lead to cloud aggregating in the absence of rain (Janssens et al., 2023; George et al., 2023). As suggested by Fig. 5 ($N_c = 1000/cm^3$, hour 38), the downward branches of these

circulations are so large that they effectively separate two cloud clusters and cause their cold pools to form at a distance from each other, thus preventing their interactions.

The high-$N_c$ behaviour corresponds to the behavior discussed by Alinaghi et al. (2024c) for simulation with $N_c = 70/cm^3$, where cold pools are characterized in analogy to squall lines in deep mesoscale convective systems (Rotunno et al., 1988; Weisman and Rotunno, 2004; Stensrud et al., 2005). In this regime, cold pools reinforce and sustain their parent clouds due

to the convergence of moist air at their fronts. Once cold pools get mature, their moist updrafts at their fronts become so strong such that they impinge on the inversion, leading to the formation of stratiform anvils with stratiform precipitation. This weakens the cold-pool-induced updraft, ultimately causing parent clouds to detach from their cold-pool children (Alinaghi et al., 2024c, their Figs. 7, 8, 11). This self-organizing behaviour of cold pools and clouds in simulations where they cannot interact is also evident in the cold-pool time series (Fig. 4cd), which exhibit an intermittent behaviour with large amplitudes

and low frequencies for both $N_c = 70$ and $1000/cm^3$.

An interesting observation is that rain and cold pools tend to develop more rapidly once the initial event has occurred. For instance, the time series shown in Fig. 4b-d demonstrates that it takes approximately 36 hours for the $N_c = 1000/cm^3$ simulation to produce rain and cold pools. However, subsequent cold-pool events occur within about 8 hours, indicating a faster formation of rain and cold pools. In this simulation, the next convective event takes place precisely where the fronts of the previous

cold-pool event had accumulated moisture, thereby expediting the development of subsequent convection. We hypothesize that cold pools in the high-$N_c$ simulation act as a "moisture memory," facilitating aggregation over shorter timescales compared to when cold pools are absent. This suggests that the moisture variance induced by cold pools compensates for the delay in rain formation associated with increased $N_c$.

As the final point in this section, we investigate how the two $N_c$-induced regimes of cold-pool self-organization dynamics

relate to the mesoscale organization of trade-cumulus cloud fields (Fig. 1, link 2). To address this, we quantify several metrics that effectively capture the variability in the mesoscale organization of cloud fields in the trades (Janssens et al., 2021). These include the domain-averaged size of cloud objects, the mean fraction of open-sky areas, the domain-mean depth of clouds, the degree of organization $I_{org}$, and the spatial standard deviation of the liquid-water path $\sigma\mathcal{L}$. The details of these metrics and their calculations are explained by Janssens et al. (2021). Following Radtke et al. (2023), we additionally consider the metric

$\Delta Q$, which quantifies the moisture aggregation at the mesoscales. This metric is calculated as the difference between the $5^{th}$ and $95^{th}$ percentiles of the mesoscale total moisture anomaly fields, derived by coarse-graining the total moisture anomaly fields as outlined by Janssens et al. (2023, see their Fig. 3).

Interestingly, the organization metrics quantifying the mean size of cloud objects and the open-sky areas, which were shown to explain most of the variability within the mesoscale organization of trade cumuli (Janssens et al., 2021), do not capture the

two distinct behaviours of cold pools at low-and high-$N_c$ regimes discussed in the context of Fig. 5 (Fig. A1). However, Fig. 6a,b shows that the spatial variance in cloud-liquid-water path $\sigma\mathcal{L}$ is strongly affected by $N_c$. First, increased $N_c$ translates





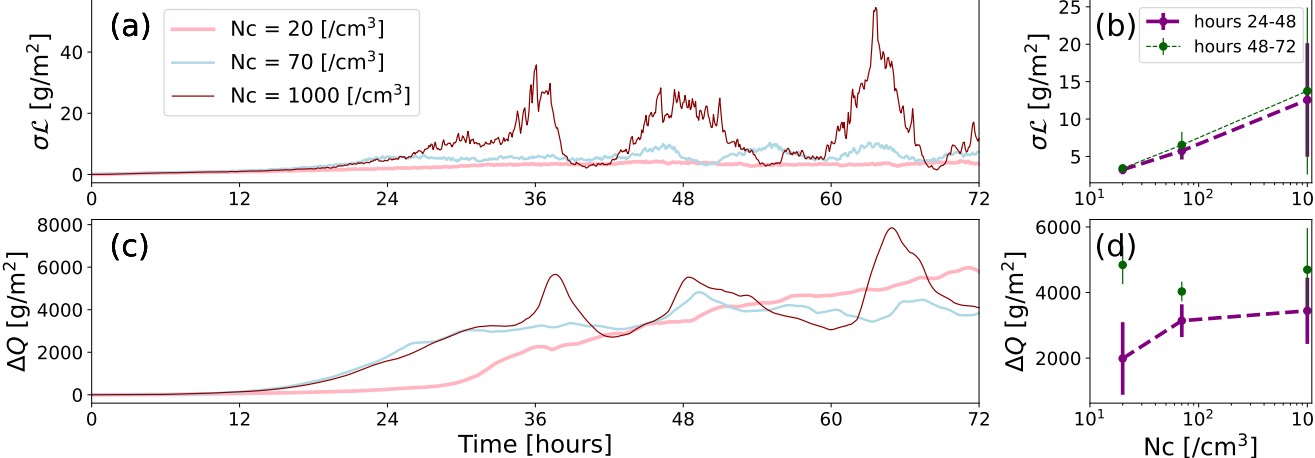

**Figure 6. Dependence of mesoscale organization of cloud fields on $N_c$.** The panels (a,c) show the time series of the spatial standard deviation of the liquid-water path $\sigma L$ and the difference between the $5^{th}$ and $95^{th}$ percentiles of the mesoscale total moisture anomaly fields $\Delta Q$ for the simulations without the diurnal cycle of insolation. Panels (b,d) show their responses to $N_c$ for hours 24-48 in purple and hours 48-72 in green. Dashed lines are added as a visual guide where a metric is sensitive to $N_c$.

into a strong intermittent behaviour in $\sigma \mathcal{L}$ evolution (Fig. 6a). Second, Fig. 6b indicates that, when averaged over 24-48 and 48-72 hour intervals, increased $N_c$ leads to greater spatial heterogeneity in liquid-water content: shallow cumuli become more aggregated in response to increased $N_c$, as visually evident in the snapshots of cloud fields shown in Figs. 3 and 5. The $\sigma \mathcal{L}$-
$N_c$ relationship suggests that a reduced number of cold pools, in response to increased $N_c$, enhances cloud aggregation. This echoes Radtke et al. (2023)'s findings, which also show that rain (auto-conversion) is less efficient in more aggregated fields of trade cumuli (Radtke et al., 2023, their Fig. 2).

With the delay in precipitation formation due to increased $N_c$, moisture is expected to continue aggregating through shallow circulations driven by latent heating from condensation in the non-precipitating cumulus layer (Bretherton and Blossey, 2017;
Janssens et al., 2023). Fig. 6c shows that this is exactly what happens in simulations with high $N_c$. In simulations with $N_c$ of 70 and 1000 /cm$^3$, the moisture aggregation metric $\Delta Q$ keeps growing until cold pools start to form, after which $\Delta Q$ stabilizes and shows an intermittent behaviour. In contrast, in the lowest case of $N_c$, the moisture does not aggregate at the mesoscales until hour 30, where cold pools start to form, after which the moisture aggregation metric $\Delta Q$ starts developing and keeps increasing until the end of the simulation. This implies that this metric of moisture aggregation at the mesoscales is intriguingly able to
encapsulate the contrasts between these distinct behaviours of cold pools at low- and high-$N_c$. Averaged over hours 24-48, $\Delta Q$ associated with the low-$N_c$ is by 50% smaller than of that of the high-$N_c$ cases (Fig. 6d). However, as the simulations progress into hours 48-72, cold pools in low-$N_c$ simulations develop into large squall lines, increasing their moisture aggregation and reducing the sensitivity of $\Delta Q$ to $N_c$ (Fig. 6d).



In summary, the evolution of cold pools and trade-cumulus fields is significantly influenced by $N_c$, exhibiting two distinct
behaviours at low and high $N_c$ (Fig. 1, links 1, 2). In simulations with $N_c = 20/cm^3$, cold pools form in close proximity, leading
to interactions and collisions that trigger the formation of clouds and due to high rain efficiency formation of new cold pools
at the collision points. This results in persistent long-lived structures resembling squall lines. Conversely, in simulations with
higher $N_c$ values (70 and $1000/cm^3$), cold pools form at greater distances, preventing interactions and resulting in intermittent
behaviour. Cold pools in such cases are short-lived structures resembling squall lines, hypothetically facilitating convection
by providing moisture anomalies at their fronts, thereby compensating for delays in subsequent cold-pool formation. This $N_c$-
driven contrast in cold-pool dynamics affects moisture and cloud-water variance, with higher $N_c$ leading to more aggregated
trade-cumulus cloud fields.

### 3.3 Diurnal cycle synchronizes the phases of cold-pool evolution across simulations with perturbed $N_c$

We have shown that the temporal evolution of cold pools, and the trade-cumulus system in general, is controlled by $N_c$ (Fig.
1, links 1, 2). Observations have shown that the evolution of trade-cumulus fields, their mesoscale organization, precipitation,
and cold pools in the trades feature diurnality (Nuijens et al., 2009; Vogel et al., 2021; Vial et al., 2021; Radtke et al., 2022).
This raises the question: to what extent does the diurnal cycle of insolation, as a time-varying external forcing, control or affect
the influence of $N_c$ on the evolution of cold pools (Fig. 1, link 3)?

To answer this question, we plot the time series of the domain-mean rain-water path $\mathcal{R}$, the number $n_{cp}$, and the size $s_{cp}$ of
cold pools in our simulations with the diurnal cycle, featuring $N_c$ values of 20, 50, 70, 100, 200, and 1000 /$cm^3$. As expected,
Figs. 7a-c show that, across all simulations, the evolution of rain and cold pools follows the diurnal cycle of radiation, peaking
around sunrise and reaching a minimum around sunset. This pattern is due to the absence of solar radiation combined with
longwave radiative cooling during the night. This strong nighttime radiative cooling destabilizes the atmosphere, stimulating
convection and leading to the formation of deeper clouds that precipitate more intensely. In contrast, during the daytime,
radiative heating from solar radiation stabilizes the atmosphere, suppressing convection and causing a notable decrease in rain,
and in the number and size of cold pools in almost all simulations. Thus, the diurnal cycle serves as an external forcing (Fig.
1, link 3) that synchronizes the periodicity and amplitude of cold-pool variability that were discussed in the context of Fig. 4
across simulations with different $N_c$. The simulated diurnal cycle in precipitation and cold pools is consistent with observations
of the trades (Nuijens et al., 2009; Vogel et al., 2021; Vial et al., 2021; Radtke et al., 2022).
Although all simulations show evolution synchronized with the diurnal cycle, the cold pools in the simulation with the lowest
$N_c$ continue to persist even during the daytime when convection suppression due to reduced net radiative cooling is at its peak.
This behaviour is notable as it indicates that the mesoscale self-organization of cold pools through collisions, as discussed in
Section 3.2, outweighs the externally imposed suppressive effect of the diurnal cycle during the day. Consequently, cold pools in
the $N_c = 20/cm^3$ simulation remain active throughout the day. However, the duration of cold-pool's persistence during the day
decreases with increasing $N_c$. Specifically, during daytime hours around 34-46, increased $N_c$ leads to an earlier disappearance
of cold pools and a more delayed formation of the next generation of cold pools, as shown in Fig. 7c. To summarize, we





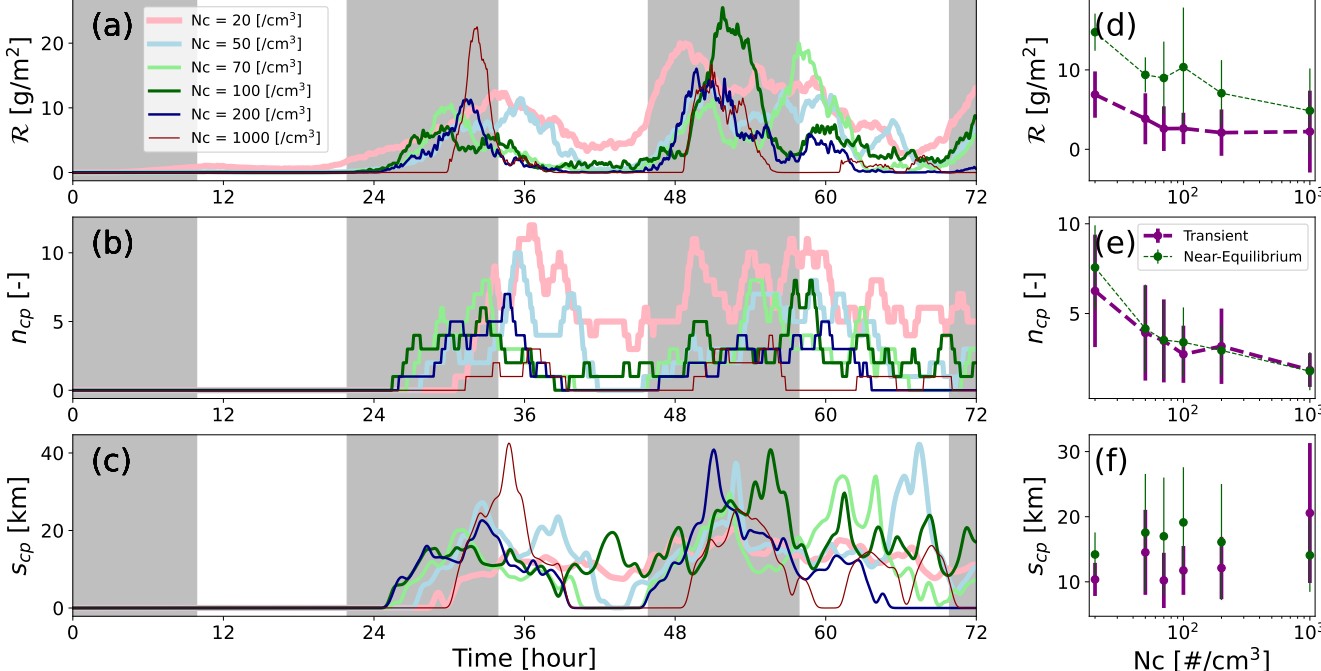

**Figure 7. Effects of the diurnal cycle of insolation on the evolution and $N_c$-response of rain and cold pools.** Panels (a-c) show time series of rain-water path $\mathcal{R}$, cold-pool number $n_{cp}$, and cold-pool size $s_{cp}$ for simulations with the diurnal cycle of solar incoming radiation. Nighttime is shown by the grey color. Panels (d-e) show the response of $\mathcal{R}$, $n_{cp}$, and $s_{cp}$ to $N_c$ during the transient (purple), and near-equilibrium (green) phases. Dashed lines are added as a visual guide where a metric is sensitive to $N_c$. Note that the reason for the cold pools in simulation $N_c = 70$ /cm$^3$ forming earlier than $N_c = 20$ /cm$^3$ is that cold pools featuring a diameter of $<5$ km are not considered to be part of the cold-pool mask. Time series of cold-pool number and size are slightly smoothed using Gaussian filtering.

find that the diurnal cycle externally synchronizes the mesoscale self-organization dynamics of cold pools, which is in turn modulated by the details of microphysics and rain formation.

The synchronization of cold-pool events by the diurnal cycle enables us to compare the responses of rain-water path $\mathcal{R}$ and cold pools to $N_c$ during the same time window across all simulations. In all simulations, except for $N_c = 20$/cm$^3$, rain and subsequently cold pools begin to form after hour 24 (Figs. 7a-c), with the evolution of cold pools following the diurnal cycle of net radiative cooling. We refer to the day starting at hour 22 as the "transient" phase and the subsequent day beginning at hour 46 as the "near-equilibrium" phase. We selected these hours based on the development of the total-water (cloud and water vapor) path in our simulations, which consistently increases until hour 48 across all simulations, after which it stabilizes and becomes time-invariant (Fig. A2).

     All simulations consistently show a higher daily mean rain-water path $\mathcal{R}$ during the near-equilibrium phase compared to the transient phase (Fig. 7d). This is because, in the near-equilibrium phase, our simulations are more developed and feature deeper boundary layers with larger total water that can develop more rain. Consistent with our results based on the last 24 hours



of simulations without the diurnal cycle (Figs. 4f-h), Figs. 7d,e illustrate that during the both transient and near-equilibrium phases, rain-water path $\mathcal{R}$ and the number of cold pools $n_{cp}$ decrease with increasing $N_c$. Also, the size of cold pools $s_{cp}$ do not show a notable change in response to $N_c$ (Fig. 7f).

### 3.4 Twomey effect primarily controls the dependence of cloud-radiative effect on $N_c$

In this section, we investigate the sensitivity of the relative cloud-radiative effect (Xie and Liu, 2013), rCRE $= f_c \cdot \mathcal{A}_c$, to $N_c$ (Fig. 1, link 4), with $f_c$ and $\mathcal{A}_c$ as cloud fraction and albedo. Assuming the plane-parallel approximation (Lacis and Hansen, 1974), cloud albedo is given by $\mathcal{A}_c = \frac{\tau}{\tau+7.7}$ with the cloud-optical thickness $\tau \approx N_c^{1/3} \mathcal{L}^{5/6}$ following Zhang et al. (2005). Therefore, we explore the response of cloud fraction $f_c$, domain-mean liquid-water path $\mathcal{L}$, mean cloud albedo $\mathcal{A}_c$ over the cloudy columns where the cloud-liquid-water path is larger than zero, and the relative cloud-radiative effect rCRE to $N_c$. These sensitivities are explored during different phases of the simulations with the diurnal cycle: non-precipitating (hours 5-15), transient (hours 22-46), and near-equilibrium (hours 46-72). For comparison, we also include results from Seifert et al. (2015) and Yamaguchi et al. (2019).

During the non- (or weakly) precipitating phase, the response of the cloud fraction $f_c$ and liquid-water path $\mathcal{L}$ to $N_c$ is negligible (Figs. 8a,b). Thus, the relative cloud-radiative effect (rCRE) is influenced by $N_c$ primarily through the cloud albedo response, or the Twomey effect (Figs. 8c,d). During both the transient and near-equilibrium phases, the cloud fraction $f_c$ decreases very slightly with increasing $N_c$, though this response is much smaller than the temporal variance of $f_c$ within each simulation (Fig. 8a). Additionally, the liquid-water path $\mathcal{L}$ shows a small increase with increased $N_c$ (Fig. 8b). However, similar to the non-precipitating phase, the Twomey effect continues to strongly control the response of rCRE to $N_c$ during both the transient and near-equilibrium phases (Figs. 8c,d).

The response of the liquid-water path in the near-equilibrium phase is consistent with the results of Yamaguchi et al. (2019) and Seifert et al. (2015). This small positive sensitivity of the liquid-water path in our simulations does not significantly affect the impact of $N_c$ on rCRE compared to the Twomey effect. Cloud fraction $f_c$ decreases with increasing $N_c$ in the simulations of Yamaguchi et al. (2019). Similarly, our simulations and those of Seifert et al. (2015) both show a decrease in $f_c$ in response to increased $N_c$. However, this decrease seems to be smaller compared with the $f_c$ response in Yamaguchi et al. (2019)'s simulations.

It is worth noting that although variations in $N_c$ modulate the number of cold pools and in turn, mesoscale self-organization of trade-cumulus fields, cloud fraction $f_c$ is very weakly affected. Our results here resonate with Janssens et al. (2024) who hypothesize that circulations associated with self-organization symmetrically distribute cloudiness at the mesoscales such that the increased cloudiness at their ascending branch is compensated by the decreased cloudiness at their descending branch. This implies that although decreased $N_c$ increases the number of cold pools, the increased cloudiness at their fronts, where convection is triggered, appears to be buffered by the decreased cloudiness at their interiors, where convection is suppressed. Future studies are encouraged to explicitly investigate this.





**Figure 8. Sensitivity of relative cloud-radiative effect to cloud-droplet number.** Panels (a-d) show the response of cloud fraction $f_c$, domain-mean liquid-water path $\mathcal{L}$, cloud albedo $\mathcal{A}_c$, and relative cloud-radiative effect rCRE to $N_c$ during the non-precipitating (dark grey), transient (purple), and near-equilibrium (green) phases. The results from Yamaguchi et al. (2019) averaged over the last 20 hours of their simulations are shown in orange. The results from Seifert et al. (2015) for their near-equilibrium state are shown in blue. Note that Yamaguchi et al. (2019); Seifert et al. (2015) do not report on the cloud albedo and rCRE in their studies. Dashed lines are added as a visual guide where a metric is sensitive to $N_c$.

### 3.5 $N_c$ induces comparable variations in cloud-radiative effect to the large-scale cloud-controlling factors

In this section, we quantify the relative importance of $N_c$ (Fig. 1, link 1) compared with the large-scale cloud-controlling factors (CCFs; Fig. 1, link 3) for driving changes in cloud-field properties and radiative effects (Fig. 1, links 2, 4). Using the data from the *Botany* dataset (Jansson et al., 2023) as well as our new simulations of this study, we employ a multivariate regression



model

$$\overline{C} \approx \sum_{i=1}^{7} \beta_i \times \widetilde{\mathrm{CCF}}_i \quad \text{with} \quad \widetilde{\mathrm{CCF}}_i := \frac{\mathrm{CCF}_i - \overline{\mathrm{CCF}}_i}{\sigma(\mathrm{CCF}_i)},$$

where, the vector $\overline{C}$ represents the mean of the metric $C \in \{\mathcal{A}_c, \mathrm{rCRE}\}$, averaged over the last 2 days (hours 12–60) for each member of the *Botany* ensemble. Each regressor $\widetilde{\mathrm{CCF}}_i$ is a vector containing the associated $\mathrm{CCF}_i$ values for the simulation members of the *Botany* ensemble. The regressors (CCFs) include sea-surface (potential) temperature ($\theta_{l0}$), near-surface wind

speed ($u_0$), moisture scale height ($h_{q_t}$), temperature lapse rate ($\Gamma$), large-scale vertical velocity variability ($w_1$), and horizontal wind shear ($u_z$). In addition, we consider $N_c$ as a $7^{th}$ regressor. For the regressor $N_c$, we use the simulations with the diurnal cycle over hours 12-60 to be consistent with the simulations of the *Botany* ensemble. All regressors are standardized by subtracting their mean $\overline{\mathrm{CCF}}$ and dividing by their standard deviation $\sigma(\mathrm{CCF})$ across the ensemble, which allows the comparison of CCFs and $N_c$ with an equal weighting. In our regression analysis, we only include simulations that develop clouds and run

for at least 48 hours. This leaves us with 80 simulations out of the initial 103 in the *Botany* ensemble. Including the simulations with the diurnal cycle from this study (six in total) and noting that the simulation with $N_c = 70/\mathrm{cm}^3$ being already part of the ensemble as the central reference simulation, our regression analysis features 85 data points in total. This means that the target value $\overline{C}$ and regressors $\widetilde{\mathrm{CCF}}_i$ of the regression model are vectors of size $85 \times 1$.

Figure 9 shows the results of the multivariate regression for cloud albedo $\mathcal{A}_c$ and rCRE. Note that the response of cloud

albedo $\mathcal{A}_c$ and rCRE to CCFs of the *Botany* ensemble has already been addressed and discussed by Janssens (2023a); Janssens et al. (2024), to which we refer for details. Figure 9a illustrates that the effect of $N_c$ on cloud albedo $\mathcal{A}_c$, known as the Twomey effect, is comparable to that of large-scale subsidence as quantified by $w_1$. Additionally, the effect of $N_c$ on trade-cumulus brightening is about 75% of the effect of horizontal wind speed, 150% of the effect of stability, and 300% of the effects of free-tropospheric humidity and vertical wind shear. Eventually, the response of rCRE to $N_c$ is statistically significant at the

95% level and accounts for about 66%, 28%, and 25% of the response of rCRE to free-tropospheric humidity, wind speed, and large-scale subsidence, respectively (Fig. 9b). Note that the sensitivity of rCRE to $N_c$ is smaller than that of cloud albedo to $N_c$ in our regression analysis, which is due to the very weak impact of $N_c$ on cloud fraction $f_c$ (Fig. 8a). Similar regression results for other cloud-field and cold-pool properties are presented in Fig. A3.

## 4  Conclusions & Outlook

Cold pools, resulting from rain-evaporation, affect the mesoscale organization of trade-cumulus fields (Zuidema et al., 2012; Seifert and Heus, 2013; Vogel et al., 2016, 2021; Alinaghi et al., 2024c). We have used an ensemble of large-domain LESs to investigate the sensitivity of mesoscale organization to cloud-droplet number concentration $N_c$ and the role of cold pools therein as conceptualized in Fig. 1.

Investigating the sensitivity of mesoscale cold-pool dynamics to microphysics (Fig. 1, link 1), we find that cold pools show

two distinct behaviours at low and high $N_c$. In low-$N_c$ cases, there are many cold pools within the simulation domain, which form in close proximity (Figs. 4c,d,g,h, 5). This allows them to interact with each other through collisions. Efficient rain





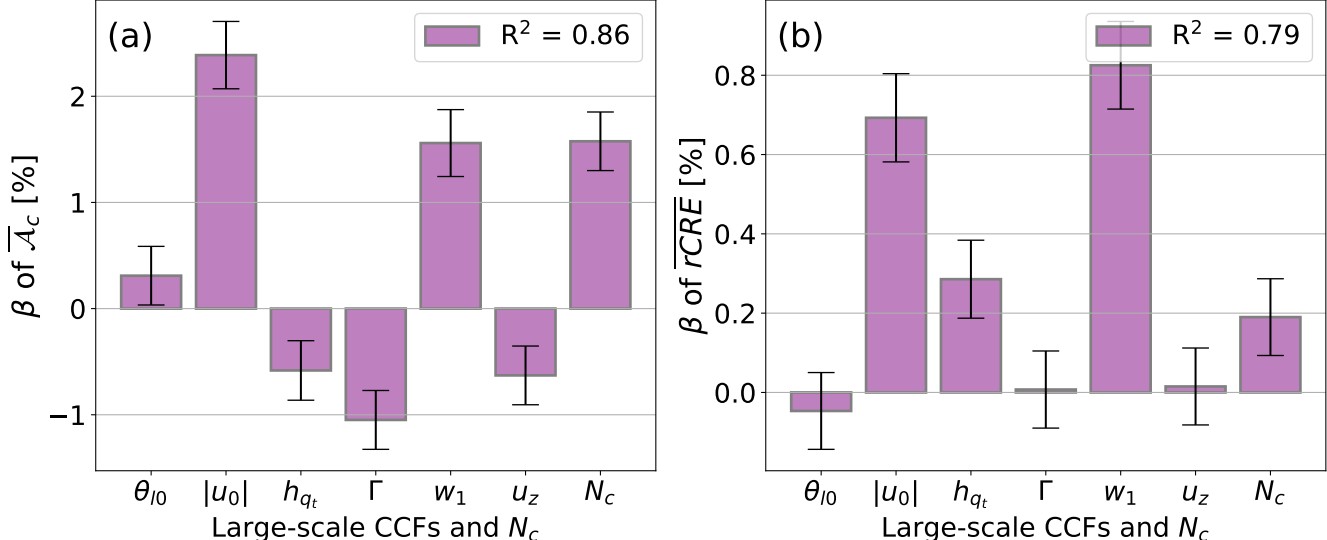

**Figure 9. Cloud-field response to large-scale cloud-controlling factors and cloud-droplet number concentration.** The standardized $\beta$ coefficients of the multiple regression analysis for (a) cloud albedo $\mathcal{A}_c$ and (b) relative cloud-radiative effect rCRE, all averaged over the last two days of the LESs of the *Botany* ensemble. The error bars show the 95% confidence interval for each regressor. The $p$-values of the $F$-statistic test of all models are smaller than $10^{-22}$.

formation then leads to the swift triggering of new cold pools at collision points. Consequently, cold pools organize into a large long-lived front with a resemblance to a squall line that perpetuates through collisions of cold pools at its leading edge. In contrast, high-$N_c$ cases feature sparsely distributed cold pools, which prevents their interaction. In this regime, cold pools

exhibit an intermittent behaviour, manifesting as small, short-lived fronts resembling squall lines that form, develop, decay, and vanish (Figs. 4c,d,g,h, 5).

For the effect of $N_c$ on the interaction between clouds and cold pools (Fig. 1, link 2), our analysis shows that increased $N_c$ suppresses the formation of cold pools (Figs. 4g,h, 5), while enhancing the self-aggregation of cloud fields (Figs. 6). In other words, by delaying precipitation formation, increased $N_c$ allows non-precipitating cumulus fields to aggregate moisture

(Fig. 6c,d) through self-reinforcing mesoscale overturning circulations (Bretherton and Blossey, 2017; Narenpitak et al., 2021; Janssens et al., 2023). We quantified this effect, which clearly influences the mesoscale organization of cloud fields, particularly by increasing the spatial variance in the cloud-liquid-water path (Fig. 6a,b). Interestingly, despite the suppression of cold pools and the boost in aggregation due to increased $N_c$ (Figs. 6a-d), the overall daily mean responses of cloud fraction and liquid-water path are notably small (Figs. 4e, 8a,b). This echoes recent findings of Janssens et al. (2024) that shallow mesoscale

circulations appear to symmetrically modulate cloudiness within trade-cumulus fields such that the increased cloudiness at their ascending branch is compensated by the decreased cloudiness at their descending branch. This suggests that although by increasing $N_c$, mesoscale organization is affected by the variations in cold pools, the increased cloudiness at the edges of cold pools appears to be buffered by the decreased cloudiness at their interiors.



For the effect of the diurnal cycle as an external forcing on the microphysical sensitivity (Fig. 1, link 3), we find that the
diurnal cycle synchronizes the self-organization dynamics of cold pools on the mesoscale across simulations with varying
$N_c$ (Fig. 7). Cold pool activity is thus controlled by both, the external forcing as well as the self-organization dynamics.
The contribution of self-organization dynamics increases with decreasing $N_c$ as showcased by the fact that cold-pool activity
survives the day-time suppression for the lowest $N_c$ case (Figs. 7b,c).

To compare the importance of microphysical and large-scale controls on the cold-pool dynamics (Fig. 1, links 1 and 3) oc-
curring on the intermediate mesoscale, we have made use of the *Cloud Botany* ensemble (Jansson et al., 2023). We demonstrate
that the Twomey effect is as significant as the primary cloud-controlling factors for the brightening of trade-cumulus fields (Fig.
9a). Despite the very small response of cloud fraction to $N_c$ (Fig. 8a), the response of the relative cloud-radiative effect to $N_c$
(Fig. 1, link 4) is about 25% as significant as the response of rCRE to horizontal wind speed and large-scale subsidence (Fig.
9b).

We have obtained these results using a prescribed cloud-droplet number as a proxy for microphysical influences. While the
more detailed microphysics scheme used by Yamaguchi et al. (2019) features a converging cloud-droplet number, Li et al.
(2015) showed that cold pools are sensitive to choices of microphysics schemes. We therefore encourage future research, such
as the cold-pool model intercomparison project (Kazil et al., 2023, CP-MIP), to focus on the sensitivity of mesoscale cold-pool
dynamics to such microphysical choices if we truly want to understand rain-evaporation, cold pools, and their relevance for
trade-cumulus fields.

Irrespective of its idealizations, this study clearly highlights that aerosol-cloud interactions are affected by processes hap-
pening at multiple spatial and temporal scales, ranging from the microscale via the mesoscale to the large scales. For shallow
cloud fields in the trades, we demonstrate that variations in the microscale can manifest themselves at the mesoscale to a degree
that is comparable to the influence of large-scale controls. We consider these findings a valuable step towards understanding
the mesoscales as a pre-requisite for constraining trade-cumulus climate feedbacks as well as trade-cumulus mediated aerosol
forcings.

*Data availability.* The *Cloud Botany* dataset is publicly accessible through the EUREC[4]A intake catalog (https://howto.eurec4a.eu/botany_
dales.html). The simulation outputs of Yamaguchi et al. (2019) are publicly available from the NOAA dataset platform (https://csl.noaa.gov/
groups/csl9/datasets/data/cloud_phys/2019-Yamaguchi-Feingold-Kazil/). The data is analyzed using Python (libraries: Numpy (Harris et al.,
2020), Xarray (Hoyer and Joseph, 2017), Pandas (Wes McKinney, 2010), Scipy (Virtanen et al., 2020), Statsmodel (Seabold and Perktold,
2010), Matplotlib (Hunter, 2007), and Seaborn (Waskom, 2021)). The basic profiles and time series associated with cloud-field properties,
cloud organization metrics, and cold-pool properties as well as a movie of simulations are publicly available by Alinaghi et al. (2024b)
and from the link https://zenodo.org/records/13868738. The coarse-graining of the total-water path anomaly fields is done using the code
(SMOCs.ipynb) by Janssens (2023b) which is publicly available (https://doi.org/10.5281/zenodo.8089287).



*Author contributions.* The concept for this study was developed by PA and FG. LES simulations were performed by FJ. PA analyzed and visualized the data, with DB assisting during the analysis. PA and FG interpreted the results with contributions from FJ and DB. PA drafted the initial manuscript, and all authors contributed to its revision and finalization. FG also supervised the work.

*Competing interests.* At least one of the (co-)authors is a member of the editorial board of Atmospheric Chemistry and Physics.

*Acknowledgements.* FG and PA acknowledge support from The Branco Weiss Fellowship - Society in Science, administered by ETH Zurich
A. FG also acknowledges support by the European Union (ERC, MesoClou, 101117462). Views and opinions expressed are however those of the author(s) only and do not necessarily reflect those of the European Union or the European Research Council Executive Agency. Neither the European Union nor the granting authority can be held responsible for them. FJ acknowledges support from the European Union's Horizon 2020 research and innovation program under grant agreement no. 820829 (CONSTRAIN project). PA and FJ thank SURF (www.surf.nl) for making the National Supercomputer Snellius accessible for running and analyzing the droplet number simulations.



**Appendix A:  Supplementary Figures**



**Figure A1. Dependence of several mesoscale cloud organization metrics on N$_c$.** The panels (a,c,e,g) show the time series of the domain-mean size of cloud objects $L_c$, the mean fraction of the open-sky areas $L_o$, the domain-mean of cloud-top height $z_t$, and the degree of organization $I_{org}$ for the simulations without the diurnal cycle of insolation. Panels (b,d,f,h) show their mean responses to N$_c$ for hours 24-48 in purple and 48-72 in green. The dashed line is shown as a visual guide where a metric is sensitive to N$_c$.



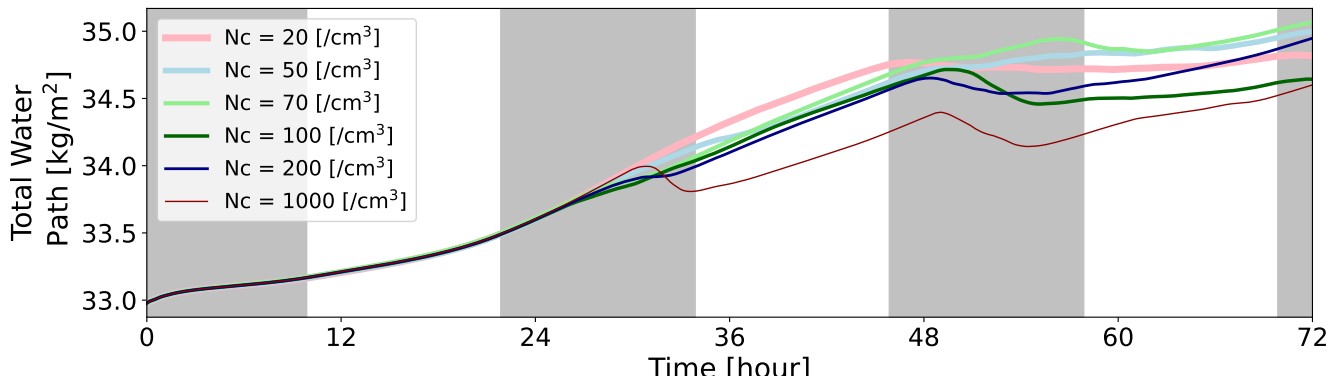

**Figure A2. Total-water path time series for several $N_c$.** The figure shows the development of the domain-mean total-water path, which is the sum of both cloud-water and water-vapor paths.

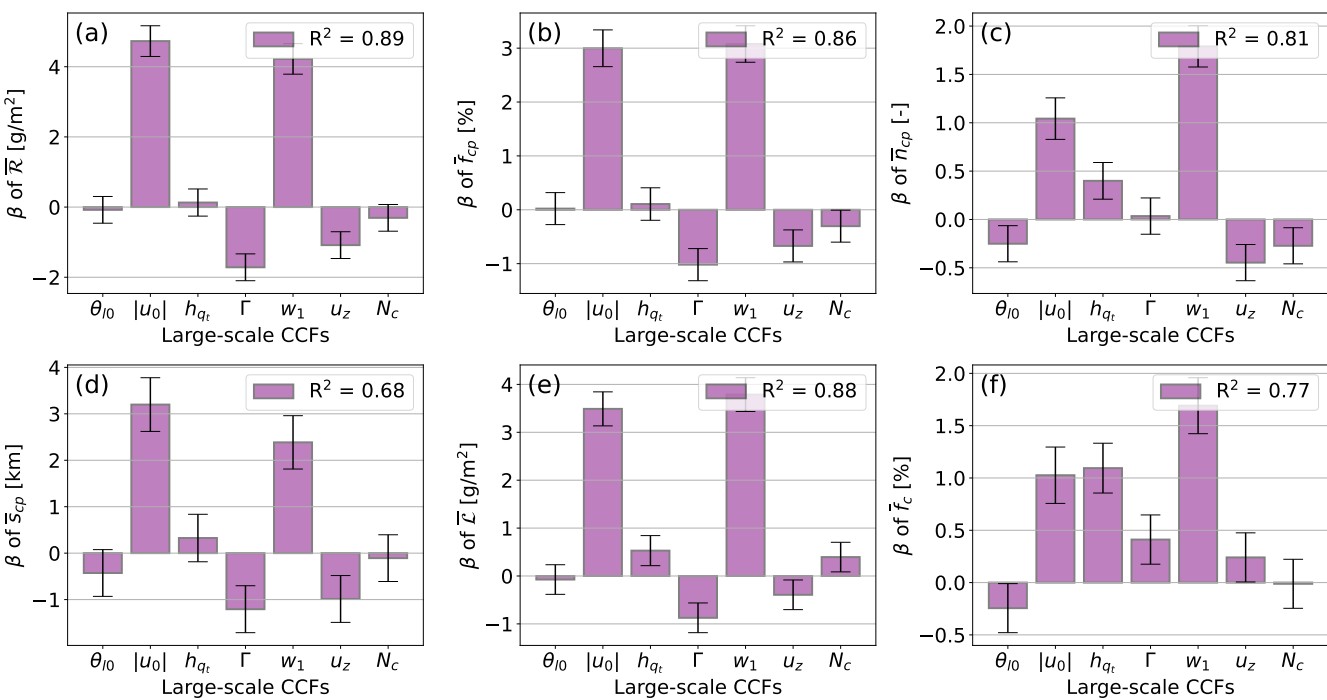

**Figure A3. Cloud-field response to large-scale cloud-controlling factors and cloud-droplet number perturbations.** The standardized $\beta$ coefficients of the multiple regression analysis for (a) rain-water path $\overline{\mathcal{R}}$, (b) cold-pool fraction $\overline{f}_{cp}$, (c) cold-pool number $\overline{n}_{cp}$, (d) cold-pool size $\overline{s}_{cp}$ (e) liquid-water path $\overline{\mathcal{L}}$, and (f) cloud fraction $\overline{f}_c$, all averaged over the last two days of the LESs of the *Botany* ensemble. The error bars show the 95% confidence interval for each regressor. The larger the distance of the confidence interval with zero, the more significant the corresponding regressor. The $p$-values of the $F$-statistic test of all models are smaller than $10^{-15}$.



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
