# Peer review of "Cold pools mediate mesoscale adjustments of trade-cumulus fields to changes in cloud-droplet number concentration"

_EGUsphere, 2024_

## Author Response (AR1)

**General response to the editor**

*Dear Raphaela Vogel,*

*We revised the text according to the comments of the reviewers. The line numbers in the text are associated with the **new tracked-changes** document, where the modifications are in purple color.*

*Thanks and best on behalf of all authors,*
*Pouriya Alinaghi*

**Reviewer #1**

This manuscript, "*Cold pools mediate mesoscale adjustments of trade-cumulus fields to changes in cloud-droplet number concentration*," presents an excellent study on the interplay between cloud-droplet number concentration (Nc) and trade-cumulus cloud fields, with a particular focus on cold pools. The study is timely, well-executed, and provides valuable insights into a topic essential for understanding cloud-climate interactions. The study identifies two distinct cold-pool regimes driven by Nc: one characterized by long-lived, organized structures formed through frequent cold-pool interactions at low Nc and another dominated by sparse, short-lived cold pools at high Nc. The study concludes that mesoscale dynamics, driven by Nc variability, significantly contribute to trade-cumulus feedback comparable to large-scale cloud-controlling factors.

In my opinion, this is a very well-written and clear manuscript that will contribute meaningfully to the field. The study is carefully designed, and the methods and results are robust. I have a few minor questions to further enhance the clarity of some concepts and results. I would like to recommend this manuscript for publication with only minor revisions.

*Dear Reviewer #1,*

*Thanks for the time you dedicated to reading our work as well as your kind words. Also thank you for your detailed comments on the quantification of cold pools. Below we have answered your comments. We also revised our text in response to your comments, which improved the quality of our manuscript.*
* * *
**L118-119:**

The sentence *"The lower boundary of hmix is determined by subtracting the difference between the upper boundary and the mode from the mode"* is somewhat unclear to me.

Would it be more correct for me to understand it as: *"The lower boundary of hmix is calculated by first determining the difference between the upper boundary and the mode, and then subtracting this difference from the mode"?*

*Thanks for your suggestion. We changed the text accordingly [now in lines 121-123].*
* * *
**Figure 4, cold pool numbers and sizes**

**Figure 4. Caption:**

*"Note that the reason for the cold pools in simulation Nc = 70 /cm³ forming earlier than Nc = 20 /cm³ is that cold pools featuring a diameter of <5 km are not considered to be part of the cold-pool mask."*

Could you clarify why an additional filtering condition could not be applied to exclude such small cold pools from the threshold? Additionally, why do such small but significantly varying cold pools appear specifically in the Nc = 70 /cm³ simulation? Understanding this would provide further insight into the dynamics of cold pools in this regime.

*We included that sentence in the caption because cold pools should follow the evolution of the rain-water path. Thus, we expect cold pools in the Nc = 20/cm³ case to begin forming earlier than those in the Nc = 70/cm³ case. So, to clarify, before hour 30, cold pools in the Nc = 20/cm³ case remain smaller than the 5-km threshold, causing them to be excluded from the time series. In contrast, cold pools in the Nc = 70/cm³ case exceed this threshold more quickly. The absence of cold pools in the Nc = 1000/cm³ case at that time is due to the delayed onset of rain formation.*

*Regardless of this threshold, the key point in our study is that once cold pools begin to form, they persist in the Nc = 20/cm³ case, whereas in cases with higher Nc (e.g., 70 or 1000/cm³), they exhibit intermittent behavior. Furthermore, the degree of intermittency—quantified by the amplitude of fluctuations in cold-pool properties—consistently*

*increases with Nc, mirroring trends observed in other cloud-related properties. All our interpretations and conclusions remain robust and are not sensitive to the specific choice of the cold-pool threshold, because none of them are based on the first hours, but rather they are based on the time period during which simulations reach their (near-)equilibrium phase.*

*In response to your comment and to avoid confusion, we changed the figure captions in Figs. 4 and 7 to "Note that there is significant precipitation in the Nc=20/cm³ case before cold pools with diameters larger than the 5-km threshold appear."*

Overall, I think the manuscript does an excellent job explaining the differences between Nc = 20 /cm³ and Nc = 1000 /cm³. However, Figure 3 suggests that the cold pools in the Nc = 50, 70, and 100 /cm³ cases exhibit the most pronounced structures and counts. Yet, these cases do not appear to receive the same level of synthesis or discussion as the extreme cases (e.g., Nc = 20 and 1000 /cm³) shown in Figure 5. Could you elaborate on the reasoning for this or include more analysis of these intermediate cases?

*Thanks for your comment and we agree with it. We did not elaborate on the Nc = 70/cm³ case in detail because it was extensively analyzed in Alinaghi et al. (2025). Following their findings, we identified this case as behaving similarly to the Nc = 1000/cm³ case, exhibiting the same intermittency, with cold pools forming far apart from one another. In response to your suggestion, we have now provided a more detailed explanation of this in lines 199-202.*
* * *
**Figure 4. Caption:**

*"Time series of cold-pool number and size are slightly smoothed using Gaussian filtering."*

Could you provide more details about the Gaussian filtering method used here? Specifically, was the same filtering applied to both cold-pool number (Figure 4c) and size (Figure 4d), and how might this smoothing influence the interpretation of the results?

*Smoothed time series are a guide to the eye only and the average response of these metrics to Nc in Figs. 4g,4h are computed based on non-smoothed data. We have clarified this in the figure caption and also added the non-smoothed time series via scatter plot in Fig. 4.*
* * *
**L155 & Figure 4d:**

The term *"domain-mean size of cold pools, Scp"* lacks clarity regarding its units and definition. In Figure 4d, the unit is shown as kilometers (km). Should this instead be km² if it represents the area, or does *"size"* specifically refer to the radius or diameter of the cold pool? Adding this clarification would help ensure consistent understanding.

*Thanks for your point. We added a detailed explanation of how we quantified the metrics number and size of cold pools at the end of the methods section [now in lines 127-130].*

*We have now added:*
*"Using the identified cold-pool mask and a clustering method, we define cold-pool objects as 2D contiguous structures within the simulation domain at each model time step. We then quantify the number of cold pools, $n_{cp}$,*

*within the domain. Additionally, we compute the domain-mean cold-pool size as $s_{cp} = \sum_{i=1}^{n_{cp}} \sqrt{A_i}/n_{cp}$, where $A_i$*

*denotes the area of each cold-pool object $i$ within the domain."*
* * *
**Figure 4:**

In addition to the mean size shown in these panels, would it be possible to include the time evolution of the 90th and 10th percentiles, or perhaps the 75th and 25th percentiles, of cold-pool sizes? This could provide a clearer picture of the size distribution over time, especially since the maximum and minimum sizes—particularly for Nc = 70 /cm³—are both extremely large and highly variable across different Nc values. Including this information might help better illustrate the variability and outlier behavior in the size distributions.

*Following your suggestion, we have quantified the time series of the 25th and 75th percentiles of cold-pool sizes, as shown below. The shaded regions represent the interquartile range (25th–75th percentile) of the cold-pool size distribution. This analysis reveals that the cold-pool size distribution broadens with decreasing Nc, which aligns with the fact that the number of cold pools is larger in cases with lower Nc (Figs. 4c, 4g).*

*While we recognize this as a valuable insight, we have chosen not to include it in the main text, as it does not directly contribute to the central message of our paper—that the intermittency of cold-pool evolution is strongly influenced by Nc.*

[Figure]

**Would Nc affect initial cold-pool field?**

Figure 5: This is a general question. The manuscript does an excellent job of explaining how, at large Nc values, the fewer cold pools and their greater distances reduce the subsequent collisions that could generate new convective events (e.g., L238, L349). However, my question concerns the initial cold-pool field: why do simulations with large Nc values already exhibit fewer and more widely spaced cold pools from the very beginning of the convective mode? Are there specific physical mechanisms at play here? For instance, could Nc directly affect precipitation evaporation rates, thereby influencing the initial cold-pool field?

*This is an interesting question. Since cold pools and rain are results of deep, aggregated clouds, we need to understand the organization of clouds just before rain formation to understand the effect of Nc on the initial cold-pool field. The reason why high Nc cases "already exhibit fewer and more widely spaced cold pools" is explained by the ΔQ metric in Fig. 6c. During the initial hours, high Nc cases—due to lower rain efficiency—develop non-precipitating clouds (see Figs. 4a, 4b), which are unstable to scale growth. This instability is driven by self-reinforcing shallow mesoscale circulations induced by latent heating from condensation (e.g., Bretherton and Blossy, 2017; Narenpitak et al., 2021; Janssens et al., 2023). As a result, both moist and dry patches continue to grow–as shown by the exponential growth in ΔQ with time in high-Nc cases–, where the growth of dry patches forces aggregated clouds atop moist patches to develop farther apart.*

*Therefore, the initial cold-pool field in high Nc cases (70 or 1000/cm³) consists of cold pools that form far from each other (see Fig. 5, Nc = 1000/cm³), as a result of a prolonged non-precipitating phase before cold-pool formation. In contrast, low Nc cases, due to higher rain efficiency, do not allow clouds to aggregate as easily as in higher Nc cases (Fig. 6c). This leads to their cold pools being more widespread within the simulation domain, forming closer together and facilitating interactions.*

*In response to your comment, we have added a more detailed explanation in lines 192-197, where we further discuss the cold-pool evolution in high Nc cases.*

**Definition of Self-Organization**

This manuscript presents an intriguing conclusion: when cold pools are suppressed, self-aggregation of cloud fields is enhanced by increased Nc (e.g., L333). However, the calculation of convective organization metrics appears challenging and nuanced. For instance, the evolution of the Iorg index shown in Figure A1 (g) and (h) does not align with this conclusion. On the other hand, I appreciate the use of the spatial standard deviation of the liquid-water path (σL) in Figure 6(a), which provides a unique perspective on cloud organization.

That said, the definition of "organization" used in this paper is somewhat unconventional, particularly since cold pools seem to play a limited role under this definition. I would suggest adding more discussion about how "organization" is defined and the physical meaning behind this choice. Such an explanation would clarify why the conclusion here—that cold pools are suppressed but the organization is enhanced—differs from the commonly understood relationship where cold pools generally increase convective organization.

*Thanks for your interesting comment. We were very careful with the use of the terms "organization" and "aggregation" throughout the manuscript.*

*Firstly, "aggregation" refers to the process during the non-precipitating phase, where moisture and clouds aggregate through self-reinforcing shallow mesoscale overturning circulations. This term is based on the work of Bretherton and Blossy (2017), Narenpitak et al. (2021), and Janssens et al. (2023). Following Radtke et al. (2023), we quantified aggregation using the ΔQ metric shown in Fig. 6c, which, although still an organization metric, specifically describes this process.*

*Secondly, once cold pools become more frequent in the field, they hinder further aggregation (as seen in Fig. 6c). However, during the non-precipitating phase for both Nc = 70 and 1000/cm³, aggregation develops freely until cold pools form. After cold-pool formation, ΔQ stabilizes and exhibits intermittency (Fig. 6c). Based on this, we concluded that cold-pool suppression enhances the self-aggregation of cloud fields.*

*These two processes—aggregation in the non-precipitating phase and cold pools in the precipitating phase—shape the mesoscale organization of clouds in ways that are interpretable. By calculating geometry-based organization metrics from Janssens et al. (2021), we found that the variability in cold pools and aggregation induced by Nc cannot be captured by these metrics. However, our process-based organization metrics, such as the size and number of cold pools, along with moisture aggregation and the spatial variance in cloud-water path, effectively capture the Nc-induced signals in our simulations.*

*In response to your comment, we have now introduced geometry-based organization metrics separately from those based on cloud-water and total moisture fields. Additionally, we emphasize that geometry-based organization metrics do not explain the Nc-induced variations in the field [now in lines 219-229].*

Additionally, I would like to request more explanation for the Nc = 70 /cm³ case. As seen in Figure 3, the clouds under this condition exhibit pronounced spatial heterogeneity (and apparent organization) from hour 30 to 50. Providing further analysis of the medium Nc case could help me—and other readers—better understand its unique characteristics and their implications.

*The Nc=70/cm³ is indeed very interesting. In response to this comment, we have now augmented lines 200-203 to highlight that our previous work analyzed and explained the self-organization behaviour of this case. It is now explained in the text that we identified this case as behaving similarly to the Nc = 1000/cm³ case, exhibiting the same intermittency, with cold pools forming far apart from one another [now in lines 199-202].*

*References for citations are provided on the last page of this document.*

**Reviewer #2**

**Summary**

Alinaghi et al. investigate the sensitivity of trade cumulus clouds and their cold pools to changes in droplet number concentration. They expand on a previously developed ensemble of high resolution, large eddy simulations (Cloud Botany) to include simulations experiencing varied, fixed droplet number concentrations (Nc= 20,50,70,100,200,1000/cc). This novel, expanded ensemble allows them to investigate how cloud behavior changes under fixed Nc and how the Nc influence compares to various cloud controlling factors (CCFs, including the diurnal cycle) that were explored in the Botany ensemble. Specifically, the analysis focuses on how cold pools, which control the organization of these clouds through their dynamics, are influenced. In addition to cold pool specifics, they analyze rain and liquid water amount and radiative properties expected to change due to the Twomey effect. Their results for rain, liquid water, and cloud amount are consistent with limited cases in the literature that had more complex microphysics. They find cold pool behaviors are different at low and high Nc concentrations, the latter leading to more spread-out cold pools and less dynamic interaction. They further find that the influence on cloud albedo is comparable to other CCFs. Overall, this work makes a strong and compelling case for needing to understand aerosol forcing in trade cumulus as well as cumulus feedback, a cutting edge concept. The manuscript is extremely well done with careful analysis that has been conducted skillfully and presented succinctly. I have a few clarifying comments and suggest that the limitations/behavioral influence due to fixed Nc should be made clearer. I recommend prompt publication after these issues are resolved as this manuscript will be a valuable contribution to the field.

*Dear Reviewer #2,*

*Thanks for the time you dedicated to reading our work, your kind words and great suggestions below. We have implemented them thoroughly and believe they improved the quality of our work and made our text more precise.*
* * *
**Major comment**

The authors are careful to focus on the Twomey effect and not fully discuss adjustments in their aci analysis. Adjustments are not possible with fixed droplet number so this focus is reasonable. However, it would be helpful to be clearer about the simplifications in your treatment of aci and how that influences the behavior of cumulus, their cold pools, and their dynamic interactions that ultimately drive their organization. Specifically, these will all be different from their behaviors in reality. The brief discussion at the end about these simplifications is good but it is worth expanding on these limitations in Section 2 when presenting the simulations and discussing their potential impact when there are key points sensitive to these assumptions. This would be particularly helpful in Section 3.4 and the discussions of cumulus lifetimes, cold pool cycling, and cold pool interactions. This added context would also help strengthen the connections between this work and current observational analysis (e.g., from EUREC4A, the basis of the Botany ensemble).

*Following your suggestion here and below in the comments, we added the limitation related to fixed Nc in the methods (lines 103-105), section 3.4 (307-308 and 321-322), and also in the conclusions (394-396).*
* * *
**Detail comments**

Intro: Compelling and very well written, synthesizing the research in the field while motivating this work well.

*Thanks!*
* * *
Intro: typo, should be "the Twomey" and "the Albrecht" effects.

*Revised accordingly [lines 24 and 28].*
* * *
Line 30-32: Consider providing a reference length and time for the "mesoscale" and "meso-timescales" mentioned.

*Added accordingly [lines 30 and 31].*
* * *
Line 75-76, throughout: This is an admirable goal. However, because not all clouds have cold pools isn't this a subset of the trade cumulus behaviors that you are looking at? Might be worth mentioning that somewhere.

*Within our study, the response of non-precipitating clouds (that do not develop cold pools) to Nc is quantified and discussed in section 3.4. Also, the mesoscale aggregation of moisture and clouds in the non-precipitating phase and its response to Nc, are explained and discussed under Fig. 6.*

*In our idealized simulations, precipitation is almost always triggered after clouds reach a certain depth. In fact, in the entire Botany ensemble, out of 87 members that develop clouds, 82 ultimately precipitate. This is consistent with observations that, in the trade-wind regime, more than 90% of the time, cloud fields are precipitating (Nuijens et al., 2009; Radtke et al., 2021). Therefore, the absence of cold pools within trade-cumulus fields is an exception rather than the rule, which is also supported by the statistics of Vogel et al. (2021).*

*Thus, the goal we mention in the introduction and the analysis throughout the manuscript are representative of the mean meteorological conditions in the trades. This is further supported by the fact that all simulations in this study are based on the mean climatology of cloud-controlling factors in the trade-wind regime, as deduced from the ERA5 data for the Botany ensemble (Jansson et al., 2023).*
* * *
Line 118-119: typo, repeated "the mode"

*We changed the text now based on the suggestion of reviewer #1 now in lines 121-123.*
* * *
Figure 4, throughout: It is great to see the similarity between your results and the cases from the literature that have used more complex microphysics, really strengthens your analysis discussion.

*Happy to hear – thanks!*
* * *
Figure 4, 6, and 7, A1 and A2: I understand your choice of colors here when you show all the Nc's. However, if you are focusing on the three (20, 70, 1000) it is a little confusing that 1000 is also red (line weight is great though, very clear, thanks). If not time consuming, would it be possible to change that line purple or something so that the colors gradate from warm to cool? Otherwise, beautiful and clear figures throughout, nice work.

*First, thanks for your kind words! Following your suggestion, we have changed the color of the Nc = 1000/cm3 case from dark red to purple throughout the plots in the main text and in the appendix.*
* * *
Line 200-202: the moisture memory hypothesis is very interesting and would be worth testing at some future point. Is their literature that supports this idea as well?

*There is indeed literature that supports the moisture memory hypothesis, although it focused on deep convection. Colin et al. (2019) demonstrated that the sources of moisture memory in deep convection mostly originate from the thermodynamic properties of the (sub-)cloud layer. They also discuss the role of cold pools in this context. We have now included a citation to their paper [line 216].*

*In future work, it will be interesting to explore how long it takes for the trade-cumulus system to regenerate cold-pool peaks, and how this timescale depends on Nc.*
* * *
Line 202-203, 240, throughout: I don't totally understand what you mean by compensating here. This seems like a key point, could you expand on what is being compensated and clarify your thinking?

*Thank you for pointing this out. By "compensating," we mean that the Nc-induced delay in rain and cold-pool formation is reduced because convection develops more quickly and easily, due to the increased moisture variance that cold pools provide.*

*To avoid any confusion, we have now replaced the verb "compensate" with "decrease," which we hope clarifies the intended meaning [line 218].*
* * *
Figure 4, 6, 7, 8, A1: Could you clarify how you are defining "sensitive to Nc"? Is there a statistical test? The error bars seem to overlap for many of the connected points… so it seems more like whether there is a trend? I found myself wondering if you could have non-monotonic sensitivity to Nc, e.g., due to some sort of buffering in cloud behavior at higher Nc concentrations…

*We had performed statistical tests for the fittings, but it's important to note that these tests are strongly influenced by the number of data points. In our case, there are a limited number of Nc points (three for Figures 4, 6, and six for Figure 7), so reporting the (large) p-values for these would be misleading, given the physical understanding of the system's response to Nc. For Figure 9, however, we have 85 data points, for which we did report p-values from the statistical tests.*

*We agree with your point here, and to avoid any confusion, we have now revised the figure captions to say: "Dashed lines are added as a visual guide where there is a trend."*
* * *
Figure 7 caption: worth mentioning the hour ranges (or marking them on the timeseries?) for quick reference of the transient and equilibrium periods.

*Yes, following your suggestion, we have now marked these phases on Figs. 7a-7c and A2 for better visualizations. Thanks for your suggestion here!*
* * *
Line 304-310:  intriguing discussion, these future studies will be very interesting.

*Thanks – we fully agree.*
* * *
Line 358-59: it would be useful to mention here, and throughout the paper when you focus on these variables, that the liquid water path and cloud fraction responses do not consider adjustments. Generally, I think you are careful about this (focusing on the Twomey effect) but worth clarifying that this is not the whole story because of the fixed Nc strategy and the behavior could differ (would you be able to generally indicate how it might differ based on what we know about adjustments?).

*We completely agree with you. We have now mentioned and emphasized on this in methods (lines 103-105), section 3.4 (307-308 and 321-322), and also in the conclusions (394-396).*

Line 375-386: Strong ending summary and discussion of some of the limitations while still highlighting the value of this paper. I would recommend discussing more about the limitations earlier on, both when you introduce the set up and when you are discussing the aci, cold pool interactions, and time evolution parts of the analysis (see other comments).

*Similar to the previous comment, we have now stressed on this limitation in our sections including methods (lines 103-105), section 3.4 (307-308 and 321-322), and also in the conclusions (394-396).*

**References**

Alinaghi, P., Siebesma, A. P., Jansson, F., Janssens, M., & Glassmeier, F. (2025). External drivers and mesoscale self‑organization of shallow cold pools in the trade‑wind regime. *Journal of Advances in Modeling Earth Systems*, *17*(1), e2024MS004540.

Bretherton, C. S., & Blossey, P. N. (2017). Understanding mesoscale aggregation of shallow cumulus convection using large‑eddy simulation. *Journal of Advances in Modeling Earth Systems*, *9*(8), 2798-2821.

Colin, M., Sherwood, S., Geoffroy, O., Bony, S., & Fuchs, D. (2019). Identifying the sources of convective memory in cloud-resolving simulations. *Journal of the Atmospheric Sciences*, *76*(3), 947-962.

Janssens, M., Vilà‑Guerau de Arellano, J., Scheffer, M., Antonissen, C., Siebesma, A. P., & Glassmeier, F. (2021). Cloud patterns in the trades have four interpretable dimensions. *Geophysical Research Letters*, *48*(5), e2020GL091001.

Janssens, M., De Arellano, J. V. G., Van Heerwaarden, C. C., De Roode, S. R., Siebesma, A. P., & Glassmeier, F. (2023). Nonprecipitating shallow cumulus convection is intrinsically unstable to length scale growth. *Journal of the Atmospheric Sciences*, *80*(3), 849-870.

Jansson, F., Janssens, M., Grönqvist, J. H., Siebesma, A. P., Glassmeier, F., Attema, J., ... & Kölling, T. (2023). Cloud Botany: Shallow Cumulus Clouds in an Ensemble of Idealized Large‑Domain Large‑Eddy Simulations of the Trades. *Journal of Advances in Modeling Earth Systems*, *15*(11), e2023MS003796.

Narenpitak, P., Kazil, J., Yamaguchi, T., Quinn, P., & Feingold, G. (2021). From sugar to flowers: A transition of shallow cumulus organization during ATOMIC. *Journal of Advances in Modeling Earth Systems*, *13*(10), e2021MS002619.

Nuijens, L., Stevens, B., & Siebesma, A. P. (2009). The environment of precipitating shallow cumulus convection. *Journal of the Atmospheric Sciences*, *66*(7), 1962-1979.

Radtke, J., Naumann, A. K., Hagen, M., & Ament, F. (2022). The relationship between precipitation and its spatial pattern in the trades observed during EUREC4A. *Quarterly Journal of the Royal Meteorological Society*, *148*(745), 1913-1928.

Radtke, J., Vogel, R., Ament, F., & Naumann, A. K. (2023). Spatial Organisation Affects the Pathway to Precipitation in Simulated Trade‑Wind Convection. *Geophysical Research Letters*, *50*(20), e2023GL103579.

Vogel, R., Konow, H., Schulz, H., & Zuidema, P. (2021). A climatology of trade-wind cumulus cold pools and their link to mesoscale cloud organization. *Atmospheric Chemistry and Physics*, *21*(21), 16609-16630.